# Identification of *mcr-1* Genes and Characterization of Resistance Mechanisms to Colistin in *Escherichia coli* Isolates from Colombian Hospitals

**DOI:** 10.3390/antibiotics12030488

**Published:** 2023-03-01

**Authors:** Elsa De La Cadena, Mateo Mahecha, Ana María Velandia, Juan Carlos García-Betancur, Laura J. Rojas, Jessica Porras, Christian Pallares, María Virginia Villegas

**Affiliations:** 1Grupo de Investigación en Resistencia Antimicrobiana y Epidemiologia Hospitalaria, Universidad El Bosque, Bogotá 110121, Colombia; 2Grupo de Investigación REMA, Facultad de Ciencias de la Salud, Programa de Bacteriología y Laboratorio Clínico, Universidad Colegio Mayor de Cundinamarca, Bogotá 110311, Colombia; 3Department of Microbiology and Molecular Biology, Case Western Reserve University, Cleveland, OH 44106-7164, USA; 4Research Service, Louis Stokes Veterans Affairs Medical Center, Cleveland, OH 44106-7164, USA; 5Comité de Infecciones y Vigilancia Epidemiológica, Clínica Imbanaco, Cali 760031, Colombia

**Keywords:** *Escherichia coli*, colistin resistance, *mcr-1*

## Abstract

We report the presence of the *mcr-1* gene among 880 *Escherichia coli* clinical isolates collected in 13 hospitals from 12 Colombian cities between 2016 and 2019. Seven (0.8%) isolates were colistin resistant (MIC ≥ 4 µg/mL). These colistin-resistant isolates were screened for the presence of the *mcr-1* gene; five carried the gene. These five isolates were subjected to whole genome sequencing (WGS) to identify additional resistomes and their ST. In addition, antimicrobial susceptibility testing revealed that all *E. coli* isolates carrying *mcr-1* were susceptible to third generation-cephalosporin and carbapenems, except one, which carried an extended-spectrum β-lactamase (CTX-M-55), along with the fosfomycin resistance encoding gene, *fosA*. WGS indicated that these isolates belonged to four distinct sequence types (ST58, ST46, ST393, and a newly described ST14315) and to phylogroups B1, A, and D. In this geographic region, the spread of *mcr-1* in *E. coli* is low and has not been inserted into high-risk clones such as ST131, which has been present in the country longer.

## 1. Introduction

The rise of multidrug-resistant (MDR), extensively drug-resistant (XDR), and pan drug-resistant (PDR) bacterial species in the clinical setting has significantly reduced available antimicrobial agents [1]. Polymyxins, particularly colistin (polymyxin E), are some of the few remaining antibiotics, and considered drugs of last resort for treating infections caused by MDR, XDR, and PDR gram-negative bacteria [2]. Unfortunately, the increasing use of colistin has led to a growing number of reports of colistin resistance worldwide, particularly in low- and mid-income countries ranging from the Mediterranean to South-East Asia, where resistance is higher than the global 10% [3].

The most common resistance mechanism to colistin in *Enterobacterales* is the structural modification of the lipid A moiety of the lipopolysaccharide (LPS) via cationic substitution [4]. The addition of cationic phosphoethanolamine (pEtN) or 4-amino-l-arabinose neutralizes the negative charge of the LPS and reduces and causes a decrease in the negative charge on the bacterial surface, which alters the binding affinity of colistin to the bacterial outer membrane [5]. Initially, this resistance was linked to mutations in chromosomally encoded genes of the two-component regulatory systems *phoP/Q, pmrA/B, crrA/B*, and the regulator *mgrB* [6,7]. Additionally, the plasmid-mediated colistin-resistance determinant *mcr-1* (mobilized colistin resistance), which encodes for a phosphoethanolamine (PEtN) transferase enzyme that adds PEtN to the lipid A [8], was originally reported in 2015 from an *Escherichia coli* isolated at a swine farm in China. Since then, more than 10 different *mcr* variants have been reported in 27 bacterial species from all six continents [9,10].

Among these variants, *mcr-1* is the most widely distributed, particularly in *E. coli*, accounting for approximately 91% of the total *mcr*-harboring isolates worldwide [11]. High prevalence rates of *mcr-1* isolates have been observed in many countries, especially in animals and farm environments, which can be caused by exposure to colistin. The spread of *mcr* genes from animals to humans challenges the clinical use of polymyxins. Recent evidence has considered animals and aquatic environments as sources and reservoirs of clinically relevant antibiotic resistance [12,13].

In Latin America, reports indicate that the prevalence of *mcr* genes in different bacterial species is low. Of a total 18,705 isolates across 48 studies, only 2.9% carried an *mcr* gene, *mcr-1* being the most frequent variant. For the particular case of Colombia, a study by Saavedra et al. found that of 513 isolates of *Enterobacterales*, only 2.3% carried *mcr-1* [11]. Additionally, Rada et al. found two isolates of *Enterobacter cloacae* complex with *mcr-9, mcr-5* was reported only in Colombia, and *mcr-3* only in Brazil [14,15,16]. In general, *mcr*-harboring isolates show high susceptibility to carbapenems, piperacillin/tazobactam, fosfomycin, and tigecycline [17]. However, the *mcr-1* gene has been found to co-occur with other resistance genes, such as various ESBLs and NDM [18]. In Colombia, *bla*_CMY-2_ and *bla*_SHV-12_ genes were identified in *E. coli* isolates harboring *mcr-1* that were resistant to colistin, and third- and fourth-generation cephalosporins. Of interest, one of these isolates was also resistant to carbapenems and monobactams [11].

The aim of this study was to characterize a set of colistin-resistant *E. coli* clinical isolates recovered from 13 hospitals in 12 Colombian cities during a four-year period (2016 to 2019). The results presented here will contribute to updating the status of colistin resistance in the country and improve the monitoring and surveillance of MDR *E. coli-mcr-1* within Colombian hospitals.

## 2. Results

### 2.1. Antimicrobial Susceptibility

A total of 880 *E. coli* clinical isolates recovered from 13 hospitals in 12 Colombian cities from 2016 to 2019 were analyzed. Of those, 7/880 (0.8%) were found to have MIC values > 4 µg/mL using the colistin broth disk elution method (CBDE) and confirmed via broth macrodilution. PCR confirmed the presence of *mcr-1* in 5/7 of these clinical isolates. Most *mcr-1* harboring isolates were susceptible to all of the other antibiotics tested, with the exception of one (28EC) that showed resistance to ciprofloxacin and a second (26EC) that showed an elevated MIC to fosfomycin (>64 µg/mL) due to the presence of the *fosA,* and exhibited an ESBL-phenotype associated with CTX-M-55 (but remained susceptible to piperacillin/tazobactam, carbapenems, and ciprofloxacin) (Table 1).

### 2.2. Whole Genome Sequencing

To further characterize the *mcr-1*-harboring *E.coli* clinical isolates, we performed WGS. Across the five isolates, the average genome size was 5.0 Mbp, with a median number of 5095 genes. In silico MLST and phylogroup analyses showed that these five isolates belonged to four different sequence types (ST) (ST58, ST46, ST393, and ST14315), and were distributed among three different phylogroups A (1), B1 (3), and D (1) (Table 2). Virulence factors play a fundamental role in conferring selective advantages and defining pathogenicity profiles in *E. coli*. Independently of their MLST, our results showed that the most frequent virulence factors among *mcr-1*-harboring isolates were: *cia*, *cvaC*, *etsC*, *fyuA*, *gad*, *hlyF*, *iroN*, *irp2*, *iss*, *iucC*, *iutA*, *ipfA*, *mchF*, *ompT*, *sitA*, *terC*, and *traT* (Table 2).

All isolates were found to carry the *mcr-1.1* variant; genetic context comparative analyses revealed that all five isolates had the *pap2* gene downstream of *mcr-1.1.* However, only isolates 26EC, 29EC, and 30EC had the mobile element IS*Apl1* upstream of *mcr-1.1* (Figure 1). Interestingly, our analyses suggested that isolates 27EC and 28EC harbor *mcr-1.1* in an IncI2 plasmid, previously associated with the mobilization of *mcr* genes. In isolate 27EC (ST58), we found the *mcr-1.1* gene in the same contig with the sequence determinants of a ~60 kb IncI2 plasmid, whereas 28EC (ST46) carried *mcr-1.1* in a contig most likely representing a ~61 kb IncI2 plasmid. Additional BLAST analyses showed a sequence similarity > 99% of these two genomic contigs with the IncI2 *mcr-1*-harboring plasmid pPK8277-3 (GenBank accession number CP016187). No other resistance genes were identified within these contigs. Unfortunately, in the remaining three isolates it was not possible to establish a direct link with a particular Inc type, as the Illumina reads assembly placed the *mcr-1.1* in a separate contig from any of the plasmid replicons identified. However, several replicon types were found (Col, IncFIA, IncFIB, IncFII, IncFIC (II), IncN, IncI1, IncHI1A, IncHI1B, IncI2, IncX4, IncI1-1, IncQ1, and IncX1), but a particular replicon type could not be co-localized with the *mcr-1.1* gene. Nonetheless, these results are consistent with previous findings suggesting that three plasmid replicon types, InHI2, IncI2, and IncX4, are primarily responsible for the global spread of *mcr-1* [19,20].

Additionally, we explored mutations in other colistin resistance-associated genes in order to identify co-determinants of colistin resistance, namely *mgrB*, *pmrA/B*, and *phoP/Q*. Our genomic analysis revealed that all five *mcr-1.1*-harboring isolates exhibited a wild type *mgrB* gene; however, three amino acid substitutions (S29G, R138H, and G144S) were found on PmrA. Of note, only the G144S substitution found on isolate 29EC was previously reported as a contributor to colistin resistance [21]. Additionally, three amino acid substitutions were detected in PmrB (H2R, D283G, and I361L), none of which were previously associated with colistin resistance. Sequence analysis of PhoP revealed one missense mutation (I44L) in four isolates, and amino acid substitutions were found on two sites, I175F and V386L. These mutations are frequently reported in *E. coli*, but their impact on colistin resistance has not been determined (Table 2) [21,22].

## 3. Discussion

The emergence and dissemination of the plasmid-mediated colistin-resistant gene *mcr-1* in *E. coli* represents a potential threat to public health. However, its real impact at the clinical level is certainly unknown [23]. In Colombia, the *mcr-1* gene was initially detected in 2016 on clinical *Enterobacterales* (*Salmonella enterica* serovar Typhimurium and *E. coli*) isolates [24]. Similarly, a previous study found twelve colistin-resistant *Enterobacterales* clinical isolates harboring *mcr-1*, including eight *E. coli* (belonging to seven different STs), which were shown to be unrelated to one another [11]. Globally, *mcr-1* and *mcr-9* are the most widely disseminated variants [25]. In Colombia, both *mcr-1* and *mcr-9* have been reported in *Enterobacterales*, including *E. coli* and *Enterobacter cloacae* complex [13,25,26]. Nonetheless, a potential limitation of our study is that no other *mcr* alleles (e.g., *mcr-2*–*mcr-10*) were screened using PCR [27]. Of note, the study carried out by Saavedra et al. found that among 513 polymyxin-resistant *Enterobacterales* only 2.3% of isolates carried *mcr-1* [11]. Our study also found a low resistance to colistin in *E. coli* (only 7/880, representing 0.8%); however, most of the resistant isolates (5/7) carried the *mcr-1.1* gene. This could potentially indicate that the prevalence of *mcr-1* in the country remains low, as previously reported by Saavedra et al. [11] and other studies in Latin America [14,28,29].

Notably, one common limitation associated with the determination of susceptibility to colistin is difficulty in screening due to the cationic nature of the compound [30]. False-resistant isolates have been reported using broth microdilution due to the use of plastic panels [31]. The CBDE method, although recommended by the CLSI guidelines, is not routinely performed in clinical microbiology laboratories, as it is laborious and time-consuming. In this study, 32 resistant isolates were identified via broth microdilution using Sensititre custom panels. The isolates were confirmed using the CBDE method and broth macrodilution (the gold-standard method); only seven isolates were colistin resistant, which highlights the difficulty in clearly establishing the real prevalence of resistance at the routinary clinical level.

Of concern, reports of clinical isolates with resistance to colistin together with resistance to other antimicrobials, such as cephalosporins and carbapenems, are increasing [32]. Notably, we did not find the co-existence of *mcr-1.1* with any carbapenemase genes, although co-carriage is being increasingly reported worldwide, including *mcr-1* with *bla*_NDM_ in *Enterobacterales* from the United States and China [33,34], and with *bla*_KPC_ in isolates from Singapore [35]. There is a high prevalence of ESBLs in *E. coli* in Colombia (up to 20%), notably of CTX-M-15 [36,37]. However, in this study, only 1/5 isolates with *mcr-1.1* showed resistance to third generation cephalosporins, due to the presence of CTX-M-55. Convergence of CTX-M and *mcr-1* genes in *E. coli* may lead to potentially widespread transmission of such resistance genes [38]. Several isolates carrying CTX-M-55 and *mcr-1* genes from animal and human origin have been reported to be increasing rapidly in geographically distinct regions [39,40]. *E. coli* ST131 is a worldwide pandemic clone, and reports have confirmed the worldwide prevalence of ST131 harboring a broad range of virulence and resistance genes on a transferable plasmid, usually associated with CTX-M-types [41]. Expansion of ST131 has been observed in both community and clinical settings, and the rapid spread of colistin-resistant ST131 is a major concern in treating MDR and XDR isolates [21]. Previous studies found a significant spread of ST131 in Colombia [36]. Fortunately, we did not find ST131 in this study, which is in agreement with results reported by other authors [13]. A wide variety of STs described in *E. coli* carriers of *mcr-1*, ST46, and ST58 reported in this study were previously found in isolates from chickens and the environment [42,43].

The *mcr-1* gene has been characterized in a variety of genomic backgrounds [44]. However, a previous study established that *mcr-1* elements derived from an initial mobilization of *mcr-1* by an IS*Apl1* transposon in the mid-2000s, which was followed by its global dissemination [45]. This mobile element also contained a downstream ORF encoding a putative PAP2 protein; the presence of these sequences could imply a role in mcr-1’s continued mobilization [45]. To date, *mcr-1* has been observed in a wider variety of plasmid types. A systematic review published by Mendes Oliveira et al. found that the plasmids carrying the *mcr-1* gene most frequently in Latin America are IncI2, IncHI1, IncHI2, IncP, IncFII, IncFIB, and IncX4 [17]. Notably, in two of our colistin-resistant *mcr-1.1*-harboring *E. coli* isolates, we identified the IncI2 replicon sequence type in the same contig as *mcr-1.1*, which might indicate its physical link within an IncI2. IncI2 plasmids are considered important drivers of the rapid mobilization and acquisition of *mcr* genes [46]. In fact, a recent study observed significant geographic clustering in the regional spread of *mcr-1*-bearing IncI2 plasmids in Asia and Brazil [47,48].

In most cases reported to date, *mcr-1* has been mobilized by the composite transposon Tn6330, containing the *mcr-1* gene, a region containing a putative open reading frame (ORF) encoding a PAP2 protein (2609 bp) flanked by two IS*Apl1* insertion sequences, although many sequences have been identified without IS*Apl1* or with just a single copy [44]. In this study, three isolates had the *mcr-1* gene embedded within an IS*Apl1*-*mcr-1*-*pap2*-IS*Apl1* cassette. Interestingly, EC27 and EC28 did not show any IS*Apl1* adjunct to the *mcr-1* gene. The plasmid sequences were identical in both isolates (27EC and 28EC), which were isolated from the same hospital during the same month in 2018; however, they had different STs and phylogroups, which could indicate plasmid dissemination. The other three isolates were collected from three different cities in 2017. Even though the number of *E. coli* isolates collected during the four years was uniform, isolates carrying *mcr-1* genes were only found in 2017 and 2018. This could be due to the low prevalence that has been reported in the country. Additionally, in 2018, Colombia prohibited the use of additives containing polymyxin or its derivatives as growth promoters in animal husbandry.

Our study faced some limitations, including that PCR detection could only be performed for *mcr-1* due to lack of controls for the other gene variants. Additionally, due to budgetary restrictions, we decided to focus on sequencing only the *mcr-1* harboring isolates, and not all colistin-resistant isolates.

## 4. Materials and Methods

### 4.1. Antimicrobial Susceptibility Testing

*E. coli* clinical isolates were collected between 2016 and 2019, as part of a prospective surveillance study by the clinical laboratories of 13 hospitals in 12 Colombian cities. All *E. coli* isolates causing infection were collected, regardless of their susceptibility profile. Isolates were sent from all participating hospitals and stored at −80 °C until further processing. Upon reception at the research lab, antimicrobial testing for susceptibility to ceftriaxone (CRO), ceftazidime (CAZ), cefepime (FEP), piperacillin/tazobactam (PTZ), ertapenem (ETP), ciprofloxacin (CIP), fosfomycin (FOS), and colistin (COL) was conducted via broth microdilution (BMD) using customized Sensititre panels (Thermo Fisher Scientific, Cleveland, OH, USA). The minimum inhibitory concentration (MIC) of ciprofloxacin was determined using the E-test™ (BioMerieux, Etoile, France). Colistin’s MIC was confirmed using the colistin broth disk elution method (CBDE) [49] and broth macrodilution using *E. coli* ATCC 25922; *mcr-1* isolate (BioSample: SAMN05806786) was included as a quality control. Results were interpreted according to the Clinical and Laboratory Standards Institute (CLSI) guidelines, 2022 [49]. Isolates with colistin MIC values > 2 µg/mL were PCR screened for *mcr-1*, as previously described [27]. 

### 4.2. Whole Genome Sequencing (WGS)

For isolates found to harbor *mcr-1* using PCR, genomic DNA was extracted using the DNeasy Blood & Tissue Kit (Qiagen, Hilden, Germany). DNA quality was verified using agarose gel electrophoresis and quantified using a Qubit 2.0 fluorometer (Invitrogen by Thermo Fisher Scientific, Life Technologies Italia, Monza, Italy). WGS was performed using a MiSeq™ platform (Illumina Inc., San Diego, CA, USA) with 250 nucleotides (nt) paired end reads to achieve a minimum coverage of 30× per nucleotide, using a MiSeq V3 flow cell. Raw reads underwent a series of steps for quality filtering, which included a general quality check and the trimming of low-quality ends; de novo assembly was performed using Spades v3.13 [50]. Assemblies were analyzed using ResFinder 4.1, PlasmidFinder 2.1, VirulenceFinder 2.0, and MLST 2.0 (according to the Achtman scheme (http://mlst.warwick.ac.uk/mlst/dbs/Ecoli, accessed on 15 July 2022) [51] from the Center for Genomic Epidemiology (http://www.genomicepidemiology.org, accessed on 15 July 2022). Furthermore, the presence of additional antibiotic resistance determinants was examined using online BLAST (https://blast.ncbi.nlm.nih.gov/Blast.cgi, accessed on 24 January 2023) [52]. Phylogroups were determined using Clermont typing (http://clermontyping.iameresearch.center/, accessed on 1 August 2022). Amino acid substitutions in additional determinants for colistin resistance PhoP/Q, PmrA/B, and MgrB were also investigated using the NCBI BLASTX tool and the genome of *E. coli* K-12 MG1655 (NC_000913.3648.1) as a reference.

## 5. Conclusions

The results presented herein suggest that colistin resistance due to the spread of *mcr-1* genes in clinical *E. coli* isolates in Colombia is still low. Fortunately, although Colombia is a country with high rates of ESBLs and KPC, *mcr-1* genes do not appear to have disseminated to high-risk clones; those *mcr-1.1* harboring *E. coli* isolates described showed high susceptibility to ertapenem, meropenem, piperacillin/tazobactam, fosfomycin, and tigecycline. Nonetheless, our study highlights the importance of routine and continuous monitoring and surveillance of colistin resistance secondary to the presence of the *mcr-1* gene in *E. coli* to trace, survey, and understand its dissemination dynamics in Colombia.

## Figures and Tables

**Figure 1 antibiotics-12-00488-f001:**
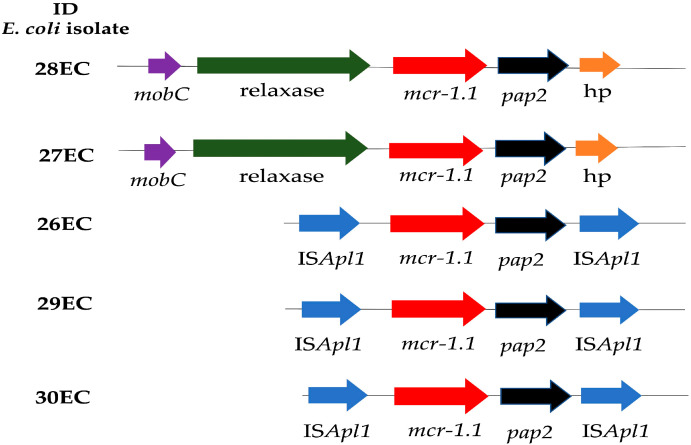
Comparative analysis of the genomic context of *mcr-1.1.* hp: hypothetic protein. Arrows represent the position and transcriptional direction of the open reading frames (ORF) regions, and *mcr-1.1* genes are indicated by red arrows.

**Table 1 antibiotics-12-00488-t001:** Demographic and minimum inhibitory concentration (MIC) for colistin and additional antimicrobial agents in *E. coli* isolates tested harboring the *mcr-1* gene.

ID	City	Isolation Date	MIC (µg/mL)								
			CRO	CTX	CAZ	FEP	TZP	ETP	CIP	FOS	COL
27EC	Bogotá	2018	<=0.5	<=0.5	<=2	<=2	<=2/4	<=0.25	0.5	<=8	**>32**
28EC	Bogotá	2018	<=0.5	<=0.5	<=2	<=2	<=2/4	<=0.25	**≥64**	<=8	**>32**
29EC	Cali	2017	<=0.5	<=0.5	<=2	<=2	<=2/4	<=0.25	0.25	<=8	**>32**
26EC	Ibagué	2017	**>4**	**4**	**8**	**8**	<=2/4	<=0.25	<0.25	**64**	**>32**
30EC	Pereira	2017	<=0.5	<=0.5	<=2	<=2	32/4	<=0.25	1	<=8	**>32**

MIC: minimum inhibitory concentration; CRO: ceftriaxone; CTX: cefotaxime; CAZ: ceftazidime; FEP: cefepime; TZP: piperacillin/tazobactam; ETP: ertapenem; CIP: ciprofloxacin; FOS: fosfomycin. MIC values in bold indicate resistance as per CLSI 2022 guidelines.

**Table 2 antibiotics-12-00488-t002:** Genomic characteristics the *mcr-1.1*-harboring *E. coli* isolates.

ID	MLST	Phylogroup	*mcr*-Variant	Resistance Genes	Plasmid Incompatibility Types	Virulence Factors
**26EC**	14315	B1	*mcr-1.1*	CTX-M-55, TEM-1B, *aph(4)-Ia, aac(3)-IV, aadA1, mdf(A), drfA1, fosA, tet(B), floR*	Col, IncFIA, IncFIB, IncFII, IncN, IncI1	*cia, cvaC, etsC, hlyF, iroN, iss, iucC, iutA, IpfA, ompT, tsh, sitA, terC, traT*
**27EC**	58	B1	*mcr-1.1*	TEM-1B, *qnrS1, Sul3, drfA15, tet(A), mdf(A), cm1A1*	Col, IncFIB, IncFIC (II), IncI2	*cea, cib, cma, cvaC, fyuA, gad, hlyF, iroN, irp2, iss, iucC, iutA, ipfA, ompT, sitA, terC, tratT*
**28EC**	46	A	*mcr-1.1*	TEM-1B, *sull2, sul3, aac(3)-Iid, aph(3′)-Ia, aadA1, tet(A), tet(M), dfrA12, mdf(A)*	Col, IncFIB, IncI2, IncX4	*celb, fyuA, gad, irp2, ompT, sitA, terC, traT*
**29EC**	393	D	*mcr-1.1*	*qnrB19, sul2, aadA1, aadA5, mdf(A), drfA17, tet(A), floR*	IncFIA, IncFII, IncHI1A, IncHI1B	*chuA, eilA, fyuA, gad, hra, iha, irp2, iss, iucC, iutA, kpsE, kpsMII, lpfA, mcmA, ompT, papA, papC, sat, sitA, terC, traT*
**30EC**	58	B1	*mcr-1.1*	TEM-1B, *qnrB19, aph(6)-Id, aadA12, aph(3″)-IIb, drfA5, sul2, floR*	InFIB, IncFII, IncI1-1, IncQ1, IncX1	*cia, cvaC, etsC, fyuA, gad, hlyF, iroN, irp2, iss, iucC, iutA, ipfA, mchF, ompT, sitA, terC, traT*

## Data Availability

All genome assemblies were deposited in the NCBI database and identified as SAMN29231639, SAMN29231636, SAMN29231637, SAMN29231638, and SAMN16793260.

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
