# Peer review of "Identification of mcr-1 Genes and Characterization of Resistance Mechanisms to Colistin in Escherichia coli Isolates from Colombian Hospitals"

_antibiotics, 2023, doi:10.3390/antibiotics12030488_

Round 1

Reviewer 1 Report

1.       In line 36 replace the term pan resistant with Pandrug-resistant (Wang, SH., Yang, Ky., Sheu, cc. Et al. The prevalence, Presentation and Outcome of Colistin Susptible only Acineobacter baumannii associated pneumonia Intensive unit: Multicenter Observational Study. SCI REP 13, 140 (2023). https://doi.org/10.1038/s41598-022-26009-0);

2.       In line 45 the email address (https://www.ncbi.nlm.nih.gov/pathoges/antimicrobialresistance/) Forwards to an articles search page, must be removed;

3.       After line 50, I suggest including articles that meet the One-Health approach (interactions between humans, animals, plants and the environment) for example:

a.       Xu L, Wan F, Fu H, Tang B, Ruan Z, Xiao Y, Luo Q. Emergence of Colistin Resistance Gene Mcr-10 in Enterobacterales Isolates Decovered from Fecal Samples of Chickens, Slaughterhouse Workers, and a Nearby Resident. Microbiol Spectr. 2022 APR 27; 10 (2): E0041822. DOI: 10.1128/Spectrum.00418-22;

b.      Cherak, Z., Lucif, L., Moussi, A., & Rolain, J.-M. (2021). Epidemiology of Mobile Colistin Resistance (MCR) Genes in Aquatic Environments. Journal of Global Antimicrobial Resistance, 27, 51–62. DOI: 10.1016/J.Jgar.2021.07.021;

4.       In line 78 remove reference number 17 from this paragraph, it must be presented in the methodology.

Author Response

Thank you very much for your comments regarding our manuscript entitled “Identification of mcr-1 genes and characterization of resistance mechanisms to colistin in Escherichia coli isolates from Colombian hospitals”. Replies to the critiques for the paper are given below in red. We hope these comments and corrections are satisfactory

  1. In line 36 replace the term pan resistant with Pandrug-resistant (Wang, SH., Yang, Ky., Sheu, cc. Et al. The prevalence, Presentation and Outcome of Colistin Susptible only Acineobacter baumanniiassociated pneumonia Intensive unit: Multicenter Observational Study. SCI REP 13, 140 (2023). https://doi.org/10.1038/s41598-022-26009-0);

Line 35. We want to thank the reviewer for his/her comments. The abbreviation was changed and we have added the reference

  1. In line 45 the email address (https://www.ncbi.nlm.nih.gov/pathoges/antimicrobialresistance/) Forwards to an articles search page, must be removed;

Line 52. Thank you for his/her observation. We have corrected this.

  1. After line 50, I suggest including articles that meet the One-Health approach (interactions between humans, animals, plants and the environment) for example:
  2. Xu L, Wan F, Fu H, Tang B, Ruan Z, Xiao Y, Luo Q. Emergence of Colistin Resistance Gene Mcr-10 in Enterobacterales Isolates Decovered from Fecal Samples of Chickens, Slaughterhouse Workers, and a Nearby Resident. Microbiol Spectr. 2022 APR 27; 10 (2): E0041822. DOI: 10.1128/Spectrum.00418-22;
  3. Cherak, Z., Lucif, L., Moussi, A., & Rolain, J.-M. (2021). Epidemiology of Mobile Colistin Resistance (MCR) Genes in Aquatic Environments. Journal of Global Antimicrobial Resistance, 27, 51–62. DOI: 10.1016/J.Jgar.2021.07.021;

Line 53. Following this great observation, we have added articles that meet the One-Health approach.

  1. In line 78 remove reference number 17 from this paragraph, it must be presented in the methodology.

 The paragraph has been changed, as recommended by the reviewer.

Reviewer 2 Report

This research focused on the identification of mcr-1 genes in Escherichia coli isolates from 13 hospitals in 12 Colombian cities from 2016 to 2019, and revealed the characterization of 5 mcr-1-positive isolates by antimicrobial susceptibility testing and whole genome sequencing. However, this research could be further improved. 

In abstract: SLV602? is MLST?

Line 73: Please delete "or". line 79, CBDE should be extended.

Table 1: The mcr-1-harboring E. coli isolates that emerged only in 2017 and 2018. Would there be more discussion? And the number of isolates collected in each year or city did not show in this article.

figure 1: ISApl1 should be a word. the same as mcr-1.1 in line 105-line 121. please carefully check.

Line 155:The prevalence of colistin-resistant isolates was at 0.8% while contrasting with the previously reported by Saavedra at 2.9% in line 58. Why indicate that the prevalence of mcr-1 is increasing?

Line 158: There was no prevalence of mcr-1 in Mendes Oliverira's report.

Line 172: There was no reference illustrating the mcr-1 with blaNDM in Enterobacteraes from the United States.

Line 205: Why were conjugation experiments not added to emphasize the dissemination of mcr-1 with or without the ISApl1 transposon?

Line 211: Were the isolates collected from the patient or otherwise? There was no ethical statement!

Line 316: "colistinresistant".

There are only four genome sequencing of  Escherichia coli in NCBI.

Author Response

Thank you very much for your comments regarding our manuscript entitled “Identification of mcr-1 genes and characterization of resistance mechanisms to colistin in Escherichia coli isolates from Colombian hospitals”. Replies to the critiques for the paper are given below in red. We hope these comments and corrections are satisfactory

This research focused on the identification of mcr-1 genes in Escherichia coli isolates from 13 hospitals in 12 Colombian cities from 2016 to 2019, and revealed the characterization of 5 mcr-1-positive isolates by antimicrobial susceptibility testing and whole genome sequencing. However, this research could be further improved. 

In abstract: SLV602? is MLST?

Line 27 – line 193. In attention to this comment, we have replaced with the newly assigned ST14315 throughout the text

Line 73: Please delete "or". line 79, CBDE should be extended.

Line 83. We have corrected this.

Table 1: The mcr-1-harboring E. coli isolates that emerged only in 2017 and 2018. Would there be more discussion? And the number of isolates collected in each year or city did not show in this article.

Line 219. We thank the reviewer for his/her constructive observation. Accordingly, we have added details regarding the isolates collected.

Figure 1: ISApl1 should be a word. the same as mcr-1.1 in line 105-line 121. please carefully check.

We adjusted the figure and corrected this typo.

Line 155:The prevalence of colistin-resistant isolates was at 0.8% while contrasting with the previously reported by Saavedra at 2.9% in line 58. Why indicate that the prevalence of mcr-1 is increasing?

Line 165. In agreement with the reviewer, we reworded that sentence to express better the idea of the sentence.

Line 158: There was no prevalence of mcr-1 in Mendes Oliverira's report.

Line 167. The reference has been changed, as recommended by the reviewer.

Line 172: There was no reference illustrating the mcr-1 with blaNDM in Enterobacteraes from the United States.

Linea 182. In attention to this comment, we have included the reference.

Line 205: Why were conjugation experiments not added to emphasize the dissemination of mcr-1 with or without the ISApl1 transposon?

Although we appreciate the reviewer’s comment, the main focus of this study was to report the isolates that harbored mcr-1 within this collection and when possible, the mobile genetic elements where they were contained. Conjugation experiments will be part of a separate study.

Line 211: Were the isolates collected from the patient or otherwise? There was no ethical statement!

Line 276. Accordingly, we have added details regarding the ethical approval information

Line 316: "colistinresistant".

Thank you. We have corrected this typo.

There are only four genome sequencing of Escherichia coli in NCBI.

Line 288. Following this observation, the five biosamples have been added. One of the sequences belongs to a bioproject previously created for another study.

Reviewer 3 Report

Title: Identification of mcr-1 genes and characterization of resistance  mechanisms to colistin in Escherichia coli isolates from Colombian hospitals

 This manuscript is thoughtful and well-written, and shares important data regarding the presence of the mcr-1 gene among 880 Escherichia coli clinical isolates from 12 Colombian hospitals between 2016 and 2019.

I  have a few minor comments: 

Line 25        In mcr-1- the hyphen should be removed.

Line 101      ‘E.coli’ should be italicized.

Line 102      ‘mcr-1’ should be italicized.

Line 187      ‘gen’ should be gene 

General comment: Authors should include the ethical approval information.

Author Response

Thank you very much for your comments regarding our manuscript entitled “Identification of mcr-1 genes and characterization of resistance mechanisms to colistin in Escherichia coli isolates from Colombian hospitals”. Replies to the critiques for the paper are given below in red. We hope these comments and corrections are satisfactory

Title: Identification of mcr-1 genes and characterization of resistance  mechanisms to colistin in Escherichia coli isolates from Colombian hospitals

 This manuscript is thoughtful and well-written, and shares important data regarding the presence of the mcr-1 gene among 880 Escherichia coli clinical isolates from 12 Colombian hospitals between 2016 and 2019.

I  have a few minor comments: 

Line 25   In mcr-1- the hyphen should be removed.

Line 101      ‘E.coli’ should be italicized.

Line 102      ‘mcr-1’ should be italicized.

Line 187      ‘gen’ should be gene 

We want to thank the reviewer for his/her positive feedback. We have corrected these minor mistakes.

General comment: Authors should include the ethical approval information.

Line 276.We thank the reviewer for his/her constructive observation. Accordingly, we have added details regarding the ethical approval information

Round 2

Reviewer 2 Report

No comment.